
# Tau-decay determination of the strong coupling

Antonio Pich[1]*

**1** IFIC, Universitat de València – CSIC, Apt. Correus 22085, E-46071 València, Spain

* antonio.pich@ific.uv.es

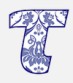
## Abstract

We review the current status of the determination of the strong coupling from tau decay. Using the most recent release of the ALEPH data, a very comprehensive phenomenological analysis has been performed, exploring all strategies previously considered in the literature and several complementary approaches. Once their actual uncertainties are properly assessed, the results from all adopted methodologies are in excellent agreement, leading to a very robust and reliable value of the strong coupling, $\alpha_s^{(n_f=3)}(m_\tau^2) = 0.328 \pm 0.013$, which implies $\alpha_s^{(n_f=5)}(M_Z^2) = 0.1197 \pm 0.0015$.

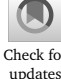
# 1 Inclusive Tau Hadronic Width

The inclusive hadronic decay width of the $\tau$ lepton is a very clean observable to determine the strong coupling with high precision [1–4]. Considering only the dominant Cabibbo-allowed decay modes, the ratio $R_{\tau,V+A} \equiv \Gamma[\tau^- \rightarrow \nu_\tau + \text{hadrons}\,(S=0)]/\Gamma[\tau^- \rightarrow \nu_\tau e^- \overline{\nu}_e]$ can be expressed through the spectral identity

$$R_{\tau,V+A} = 12\pi |V_{ud}|^2 S_{\text{EW}} \int_0^{m_\tau^2} \frac{ds}{m_\tau^2} \left(1 - \frac{s}{m_\tau^2}\right)^2 \left[\left(1 + 2\frac{s}{m_\tau^2}\right) \text{Im}\,\Pi^{(1)}_{V+A}(s) + \text{Im}\,\Pi^{(0)}_{V+A}(s)\right], \quad (1)$$

where $\Pi^{(J)}_{\mathcal{J}}(s)$ ($\mathcal{J} = V, A; J = 0, 1$) are the two-point correlation functions for the vector $V^\mu = \overline{u}\gamma^\mu d$ and axial-vector $A^\mu = \overline{u}\gamma^\mu \gamma_5 d$ colour-singlet light-quark charged currents:

$$i \int d^4x \, e^{iqx} \langle 0| T[\mathcal{J}^\mu(x)\mathcal{J}^{\nu\dagger}(0)]|0\rangle = (-g^{\mu\nu}q^2 + q^\mu q^\nu) \Pi^{(1)}_{\mathcal{J}}(q^2) + q^\mu q^\nu \Pi^{(0)}_{\mathcal{J}}(q^2). \quad (2)$$

The factor $S_{\text{EW}} = 1.0201 \pm 0.0003$ incorporates the electroweak corrections [5–7]. For massless quarks, $\Pi^{(0)}_V(s) = 0$ while $s\,\Pi^{(0)}_A(s)$ is a known constant, generated by the pion pole contribution, that cancels in $\Pi^{(0+1)}_A(s)$.

The measured invariant-mass distribution of the final hadrons determines the spectral functions $\rho_{\mathcal{J}}(s) \equiv \frac{1}{\pi} \text{Im}\,\Pi^{(0+1)}_{\mathcal{J}}(s)$, shown in Fig. 1 [8]. Using the analyticity properties of the corre-

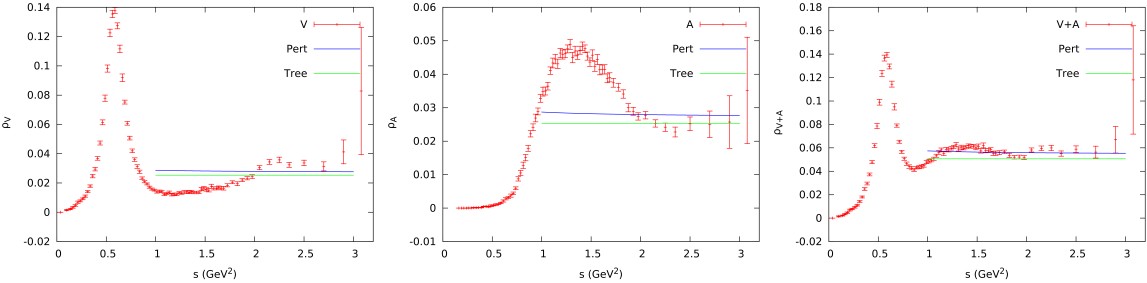

Figure 1: ALEPH spectral functions for the $V$, $A$ and $V + A$ channels [8].

lators, this experimental information can be related with theoretical QCD predictions through moments of the type [4,9]

$$A^\omega_{\mathcal{J}}(s_0) \equiv \int_{s_{\text{th}}}^{s_0} \frac{ds}{s_0} \, \omega(s) \, \text{Im}\,\Pi^{(0+1)}_{\mathcal{J}}(s) = \frac{i}{2} \oint_{|s|=s_0} \frac{ds}{s_0} \, \omega(s) \, \Pi^{(0+1)}_{\mathcal{J}}(s), \quad (3)$$

where $\omega(s)$ is any weight function analytic in $|s| \le s_0$, $s_{\text{th}}$ is the hadronic mass-squared threshold, and the complex integral in the right-hand side runs counter-clockwise around the circle $|s| = s_0$. For large-enough values of $s_0$, this contour integral can be predicted as an expansion in inverse powers of $s_0$, using the operator product expansion (OPE) of the current correlators:

$$\Pi^{(0+1)}_{\mathcal{J}}(s)\Big|^{\text{OPE}} = \sum_D \frac{1}{(-s)^{D/2}} \sum_{\dim \mathcal{O}=D} C_{D,\mathcal{J}}(-s,\mu) \langle \mathcal{O}(\mu)\rangle \equiv \sum_D \frac{\mathcal{O}_{D,\mathcal{J}}}{(-s)^{D/2}}. \quad (4)$$

Differences between the physical values of the integrated moments $A^\omega_{\mathcal{J}}(s_0)$ and their OPE approximations are known as quark-hadron duality violations. They are very efficiently minimized by taking "pinched" weight functions which vanish at $s = s_0$, suppressing in this way the contributions from the region near the real axis where the OPE is not valid [4,9].

## 2   Perturbative Contribution

The perturbative contribution (*i.e.*, the $D = 0$ term in the OPE) dominates the moments $A_{\mathcal{J}}^{\omega}(s_0)$, when $s_0 \sim \mathcal{O}(m_{\tau}^2)$. The chiral symmetry of QCD guarantees that the vector and the axial-vector perturbative correlators are identical for massless quarks. Their perturbative calculation is conveniently expressed in terms of the Adler function

$$D(s) \equiv -s \frac{d\,\Pi_V^{(0+1)}(s)}{ds} = \frac{1}{4\pi^2} \sum_{n=0} \tilde{K}_n(\xi) \left(\frac{\alpha_s(-\xi^2 s)}{\pi}\right)^n. \tag{5}$$

The coefficients $K_n \equiv \tilde{K}_n(\xi = 1)$ are known up to $n \leq 4$. For $n_f = 3$ flavours, they have the values: $K_0 = K_1 = 1$, $K_2 = 1.63982$, $K_3^{\overline{\text{MS}}} = 6.37101$ and $K_4^{\overline{\text{MS}}} = 49.07570$ [10]. $D(s)$ satisfies an homogeneous renormalization-group equation, which determines the corresponding scale-dependent parameters $\tilde{K}_n(\xi)$ [11–13].

The perturbative contribution to $A_{\mathcal{J}}^{\omega}(s_0)$ can be written as a series involving the Adler coefficients $\tilde{K}_n(\xi)$ multiplied by contour integrals that only depend on $\alpha_s(\xi^2 s_0)$:

$$A^{\omega,P}(s_0) = -\frac{1}{8\pi^2 s_0} \sum_{n=0} \tilde{K}_n(\xi) \int_{-\pi}^{\pi} d\varphi \, \left[W(-s_0\,e^{i\varphi}) - W(s_0)\right] \left(\frac{\alpha_s(\xi^2 s_0\,e^{i\varphi})}{\pi}\right)^n, \tag{6}$$

with $W(s) \equiv \int_0^s ds'\,\omega(s')$. These integrals can be computed with high accuracy solving the $\beta$-function equation, up to unknown $\beta_{n>5}$ contributions. One gets in this way a *contour-improved perturbation theory* (CIPT) series [11,14], which sums big running corrections arising at large values of $\varphi$, is stable under changes of the renormalization scale $\xi$ and has a good perturbative convergence. A naive truncation of the integrals, to a fixed order in $\alpha_s(\xi^2 s_0)$ (*fixed-order perturbation theory*, FOPT) [4], results instead in a series with slower convergence and a larger dependence on $\xi$.

The theoretical prediction for the ratio $R_{\tau,V+A}$ is given by [4]

$$R_{\tau,V+A} = N_C\,|V_{ud}|^2\,S_{\text{EW}}\,\{1 + \delta_{\text{P}} + \delta_{\text{NP}}\}, \tag{7}$$

where $\delta_{\text{NP}}$ contains the small non-perturbative contribution, plus negligible quark-mass corrections smaller than $10^{-4}$. The dominant perturbative component takes the form [11]

$$\delta_{\text{P}} = \sum_{n=1} K_n A^{(n)}(\alpha_s) = a_{\tau} + 5.2023\,a_{\tau}^2 + 26.3659\,a_{\tau}^3 + 127.079\,a_{\tau}^4 + (K_5 + 307.78)\,a_{\tau}^5 + \cdots, \tag{8}$$

with $a_{\tau} = \alpha_s(m_{\tau}^2)/\pi$. The functions $A^{(n)}(\alpha_s) = a_{\tau}^n + \mathcal{O}(a_{\tau}^{n+1})$ are the contour-integrals in Eq. (6) for the weight function $\omega_{R_{\tau}}(x) = (1-x)^2(1+2x)$, with $x = s/s_0$, $s_0 = m_{\tau}^2$ and $\xi = 1$. Their numerical values up to $n = 4$ are displayed in Table 1, for different loop approximations, exhibiting a very good perturbative convergence. Their FOPT expansion in Eq. (8) generates instead a very slowly-converging series with coefficients much larger than the original $K_n$ factors.

The perturbative error associated with the unknown higher-order corrections to the Adler function is the dominant theoretical uncertainty in the determination of the strong coupling. For a fixed value of $\alpha_s$, FOPT gives a larger perturbative contribution than CIPT and, therefore, results in a smaller fitted value of $\alpha_s(m_{\tau}^2)$. In our numerical analyses we will consider both schemes, taking the conservative range $K_5 = 275 \pm 400$ and varying the renormalization scale in the interval $\xi^2 \in (0.5, 2)$.

Table 1: Exact results for $A^{(n)}(\alpha_s)$ ($n \leq 4$) at different $\beta$-function approximations, and corresponding values of $\delta_{\mathrm{P}} = \sum_{n=1}^{4} K_n A^{(n)}(\alpha_s)$, for $a_\tau \equiv \alpha_s(m_\tau^2)/\pi = 0.11$. The last row shows the FOPT estimates at $\mathcal{O}(a_\tau^4)$, which overestimate $\delta_P$ by 11% [15].

|  | $A^{(1)}(\alpha_s)$ | $A^{(2)}(\alpha_s)$ | $A^{(3)}(\alpha_s)$ | $A^{(4)}(\alpha_s)$ | $\delta_{\mathrm{P}}$ |
|---|---|---|---|---|---|
| $\beta_{n>1} = 0$ | 0.14828 | 0.01925 | 0.00225 | 0.00024 | 0.20578 |
| $\beta_{n>2} = 0$ | 0.15103 | 0.01905 | 0.00209 | 0.00020 | 0.20537 |
| $\beta_{n>3} = 0$ | 0.15093 | 0.01882 | 0.00202 | 0.00019 | 0.20389 |
| $\beta_{n>4} = 0$ | 0.15058 | 0.01865 | 0.00198 | 0.00018 | 0.20273 |
| $\mathcal{O}(a_\tau^4)$ | 0.16115 | 0.02431 | 0.00290 | 0.00015 | 0.22665 |

## 3  Sensitivity to the Strong Coupling

The high sensitivity of $R_{\tau,V+A}$ to $\alpha_s(m_\tau^2)$ follows from a combination of several facts [4]:

1. The perturbative contribution is known to $\mathcal{O}(\alpha_s^4)$. Since $\alpha_s(m_\tau^2) \sim 0.33$ is sizeable, $\delta_{\mathrm{P}}$ amounts to a quite large 20% effect, making $R_{\tau,V+A}$ more sensitive to the strong coupling than higher-energy observables.

2. The OPE can be safely used at $s_0 = m_\tau^2$. The integrand in Eq. (1) involves a double zero at $s = m_\tau^2$, heavily suppressing the contribution from the region near the real axis, where the OPE is not valid, to the corresponding contour integral.

3. The relevant $\Pi^{(0+1)}(s)$ contribution is weighted with $\omega_{R_\tau}(x) = (1-x)^2(1+2x) = 1-3x^2+2x^3$. Cauchy's theorem implies then that the contour integral is only sensitive to OPE corrections with $D = 6$ and 8, which are strongly suppressed by the corresponding powers of the $\tau$ mass. There is in addition a strong cancellation between the the vector and axial-vector power corrections, which have opposite signs. This cancellation was theoretically predicted for the $D = 6$ contributions [4], but the $\tau$-data analyses show that it is also operative in the $D = 8$ terms [8, 16].

4. Fig. 1 shows that the inclusive $V + A$ spectral distribution is very flat. The opening of high-multiplicity hadronic thresholds dilutes very soon the prominent $\rho(2\pi)$ and $a_1(3\pi)$ resonance peaks. The data approaches very fast the perturbative QCD predictions that seem to work even at surprisingly low values of $s \sim 1.2\,\mathrm{GeV}^2$.

The small correction $\delta_{\mathrm{NP}}$ can be determined from the hadronic $\tau$ data, analysing spectral moments more sensitive to power corrections [9]. The detailed studies performed by ALEPH [17–21], CLEO [22] and OPAL [23] have confirmed that non-perturbative contributions are below 1%, i.e., smaller than the perturbative uncertainties. The latest and most precise experimental determination of the strong coupling, performed with the ALEPH data, gives $\delta_{\mathrm{NP}} = -0.0064 \pm 0.0013$ and $\alpha_s^{(n_f=3)}(m_\tau^2) = 0.332 \pm 0.005_{\mathrm{exp}} \pm 0.011_{\mathrm{th}}$ [8]. The second uncertainty takes into account the different central values obtained with the FOPT (0.324) and CIPT (0.341) prescriptions, adding quadratically half their difference as an additional systematic error. Taking as input the small $\delta_{\mathrm{NP}}$ correction extracted from the ALEPH analysis, $\alpha_s$ can be also determined directly from the total $\tau$ hadronic width (and/or lifetime); this gives $\alpha_s^{(n_f=3)}(m_\tau^2) = 0.331 \pm 0.013$ (FOPT + CIPT) [15], in perfect agreement with the ALEPH result.

Table 2: Determinations of $\alpha_s^{(n_f=3)}(m_\tau^2)$ from $\tau$ decay data, in the $V+A$ channel [16].

| Method | $\alpha_s^{(n_f=3)}(m_\tau^2)$ | | |
|---|---|---|---|
| | CIPT | FOPT | Average |
| $\omega_{kl}(x)$ weights | $0.339^{+0.019}_{-0.017}$ | $0.319^{+0.017}_{-0.015}$ | $0.329^{+0.020}_{-0.018}$ |
| $\hat{\omega}_{kl}(x)$ weights | $0.338^{+0.014}_{-0.012}$ | $0.319^{+0.013}_{-0.010}$ | $0.329^{+0.016}_{-0.014}$ |
| $\omega^{(2,m)}(x)$ weights | $0.336^{+0.018}_{-0.016}$ | $0.317^{+0.015}_{-0.013}$ | $0.326^{+0.018}_{-0.016}$ |
| $s_0$ dependence | $0.335 \pm 0.014$ | $0.323 \pm 0.012$ | $0.329 \pm 0.013$ |
| $\omega_a^{(1,m)}(x)$ weights | $0.328^{+0.014}_{-0.013}$ | $0.318^{+0.015}_{-0.012}$ | $0.323^{+0.015}_{-0.013}$ |
| Average | $0.335 \pm 0.013$ | $0.320 \pm 0.012$ | $0.328 \pm 0.013$ |

## 4 Updated Determination of $\alpha_s(m_\tau^2)$

The previous experimental analyses have been criticized in Ref. [24], where slightly smaller (10%) values of $\alpha_s(m_\tau^2)$ are obtained with a different method that maximises the role of non-perturbative effects. The uncertainties of this determination are, however, largely underestimated. We relegate to the appendix a brief description of the conceptual and numerical flaws that question the claimed accuracy. In view of the triggered controversy, we have performed a complete re-analysis of the updated ALEPH data, exploring a large variety of methodologies that include all previously considered methods (also the one advocated in Ref. [24]), trying to uncover their potential hidden weaknesses and testing the stability of their results under slight variations of the assumed inputs [16].

The most reliable determinations, extracted from the $V+A$ hadronic distribution, are summarized in Table 2. The systematic difference between the CIPT and FOPT results is clearly manifested in the table. The values obtained from both procedures have been conservatively combined, following the same prescription than Ref. [8]. In addition to the perturbative errors, estimated as indicated in section 2, all quoted results include as additional theoretical uncertainty their variations under various modifications of the fit procedures. Similar (and consistent) results, although with larger uncertainties, are obtained from the separate $V$ and $A$ distributions.

The first three determinations are based on the (at least double-pinched) weights

$$
\begin{aligned}
\omega_{kl}(x) &= (1-x^2)^{2+k} x^l (1+2x), & (k,l) &= \{(0,0),(1,0),(1,1),(1,2),(1,3)\}, \\
\hat{\omega}_{kl}(x) &= (1-x^2)^{2+k} x^l, & (k,l) &= \{(0,0),(1,0),(1,1),(1,2),(1,3)\}, \\
\omega^{(2,m)}(x) &= 1-(m+2) x^{m+1} + (m+1) x^{m+2}, & 1 &\leq m \leq 5, \quad (9)
\end{aligned}
$$

with $x = s/s_0$. Taking $s_0 = m_\tau^2$, the functions $\omega_{kl}(x)$ include the phase-space and spin-1 factors in Eq. (1), allowing for a direct use of the measured spectral distribution and the precise determination of $R_{\tau,V+A}$. Following the ALEPH analysis, we have performed a global fit of $\alpha_s(m_\tau^2)$, the gluon condensate, $\mathcal{O}_6$ and $\mathcal{O}_8$ with the five moments indicated. The sensitivity of $\alpha_s$ to neglected higher-order condensates (the highest moment involves $D \leq 16$ power corrections) has been estimated through a second fit including $\mathcal{O}_{10}$ and the variation has been included as an additional uncertainty. The final results, shown in the first line of Table 2, nicely agree with Ref. [8]. A similar fit with the modified $\hat{\omega}_{kl}(x)$ weights, that eliminate from every moment the highest-$D$ condensate contribution, gives the results in the second line of the table, in perfect agreement with the previous fit. Thus, the sensitivity to power corrections is quite small, which gets reflected in rather large uncertainties of the fitted condensates.

The optimized moments $\omega^{(2,m)}(x)$, that only receive condensate contributions from $\mathcal{O}_{2(m+2)}$

and $\mathcal{O}_{2(m+3)}$, lead to the results in the third line of Table 2. We have first made a combined fit of five different moments ($1 \leq m \leq 5$), assuming $\mathcal{O}_{12} = \mathcal{O}_{14} = \mathcal{O}_{16} = 0$, and a second fit including $\mathcal{O}_{12}$ has been used to asses the induced uncertainty from missing power corrections. The agreement with the previous $\omega_{kl}(x)$ and $\hat{\omega}_{kl}(x)$ fits is excellent. Similar results (not shown in the table) are obtained from a global fit to four moments, based on the weights $\omega^{(n,0)}(x) = (1-x)^n$, with $0 \leq n \leq 3$.

The strong coupling can be determined with a single moment, provided all non-perturbative corrections are neglected. Obviously, this cannot be used to extract an accurate value of $\alpha_s$, but it constitutes a very interesting exercise to assess the minor numerical role of non-perturbative effects. Thirteen separate extractions of the strong coupling have been made in Ref. [16], with the weights $\omega^{(0,0)}(x)$, $\omega^{(1,m)}(x) = 1 - x^{m+1}$ and $\omega^{(2,m)}(x)$ ($0 \leq m \leq 5$). Power corrections are absent with the first weight, but the corresponding moment is very exposed to violations of quark-hadron duality. Moments with the second type of weights are only sensitive to a single condensate, $\mathcal{O}_{2(m+2)}$, while both $\mathcal{O}_{2(m+2)}$ and $\mathcal{O}_{2(m+3)}$ contribute with the third type. In spite of this large disparity of neglected non-perturbative corrections, the thirteen determinations turn out to be in good agreement with the results given in Table 2, exhibiting an amazing stability of the fitted value of $\alpha_s(m_\tau^2)$.

## 5 Sensitivity to $s_0$

Non-perturbative contributions leave their imprint in a distinctive dependence on $s_0$ of the different moments. The spectral integrals $A^{(1,m)}(s_0)$, built with $\omega^{(1,m)}(x)$ weights, get a power correction that scales as $1/s_0^{m+2}$, while using the weight functions $\omega^{(2,m)}(x)$ one gets $1/s_0^{m+2}$ and $1/s_0^{m+3}$ power corrections on the corresponding $A^{(2,m)}(s_0)$ moments. The $s_0$ dependence of the experimental moments $A^{(1,0)}(s_0)$ and $A^{(2,0)}(s_0)$ is displayed in Fig. 2, for the vector, axial-vector and $\frac{1}{2}(V+A)$ distributions, together with their predicted perturbative values with $\alpha_s(m_\tau^2) = 0.329^{+0.020}_{-0.018}$, i.e. neglecting all non-perturbative effects.

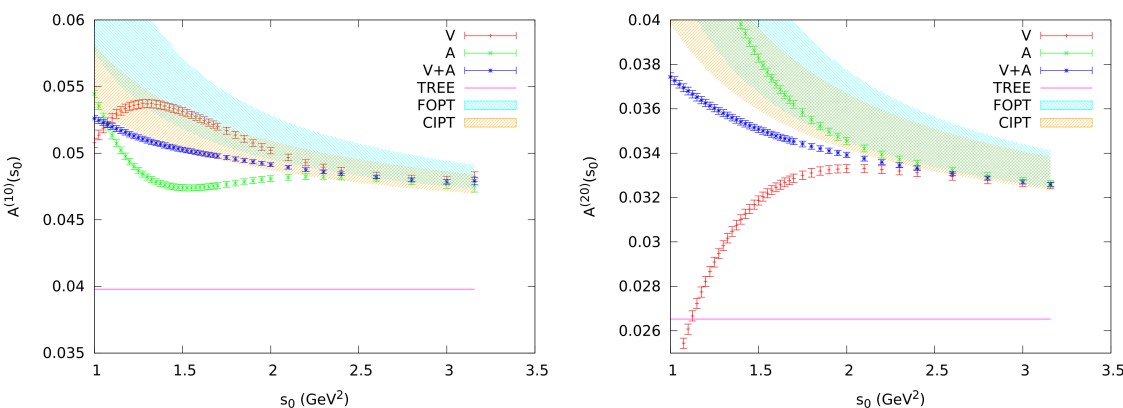

Figure 2: Dependence on $s_0$ of the experimental moments $A^{(1,0)}(s_0)$ (left) and $A^{(2,0)}(s_0)$ (right), for the $V$ (red), $A$ (green) and $\frac{1}{2}(V+A)$ (blue) channels. The orange and light-blue regions are the CIPT and FOPT perturbative predictions for $\alpha_s(m_\tau^2) = 0.329^{+0.020}_{-0.018}$ [16].

Despite being only protected by a single pinch factor, the measured $A^{(1,0)}(s_0)$ agrees with its pure perturbative prediction in all channels ($V$, $A$ and $V+A$), following the CIPT central values above $s_0 \sim 2$ GeV$^2$. This moment can only get power corrections from $\mathcal{O}_4$ that are

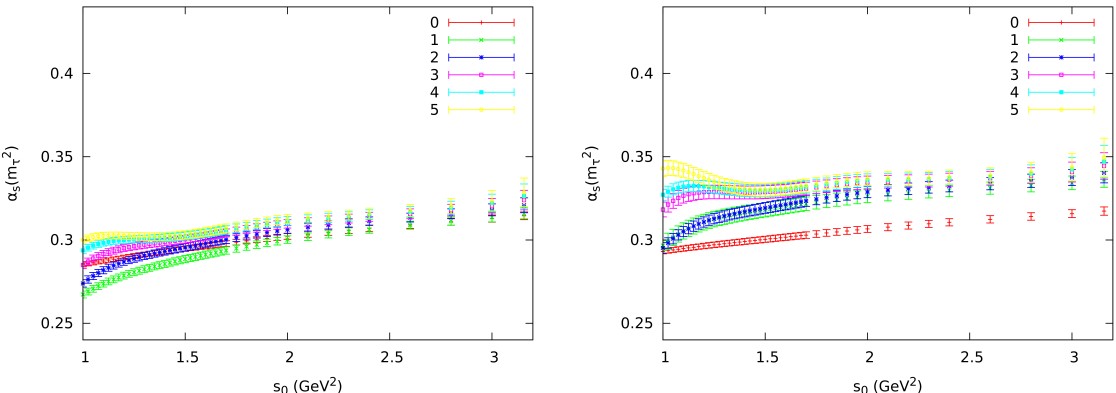

Figure 3: $\alpha_s(m_\tau^2)$ determinations with FOPT (left) and CIPT (right), at different values of $s_0$ and from different $(V+A)\, A^{(2,m)}(s_0)$ moments ($0 \le m \le 5$), ignoring all non-perturbative effects [16]. Only experimental uncertainties are displayed.

approximately equal for the $V$ and $A$ correlators, but they turn out to be too small to become visible within the much larger perturbative errors indicated by the broad shaded areas. The $V$ and $A$ experimental curves split at smaller $s_0$ values, signaling the presence of duality violations; however, these effects compensate to a large extent in $V+A$, leaving an astonishingly flat distribution that remains within the $1\sigma$ perturbative range even at $s_0 \sim 1$ GeV. A similar behaviour is observed for the non-protected moment $A^{(0,0)}(s_0)$, which does not receive any OPE corrections [16].

The experimental $A^{(2,0)}(s_0)$ curves suggest the presence of a power correction with different signs for $V$ and $A$, which largely cancels in $V+A$. This nicely matches the behaviour expected from the $\mathcal{O}_{6,V/A}$ contribution. However, the numerical size of this correction seems to be tiny at $s_0 \sim m_\tau^2$ because the $V$, $A$ and $V+A$ distributions join above $s_0 \sim 2.2$ GeV$^2$.

Six different determinations of the strong coupling from $A^{(2,m)}(s_0)$ ($0 \le m \le 5$) moments, neglecting all non-perturbative corrections, are shown in Fig. 3 as function of $s_0$. Similar results have been also obtained from seven $A^{(1,m)}(s_0)$ integrals ($0 \le m \le 6$) [16]. The missing non-perturbative corrections to all these moments are very different, spanning a broad variety of inverse powers of $s_0$. However, this diversity of power corrections does not show up in the figure: all curves cluster and display a similar $s_0$ dependence, suggesting very small power corrections for $V+A$. Notice that only experimental errors have been shown in the figure. The fluctuations of the different determinations remain always within the much larger perturbative uncertainties indicated in Fig. 2.

From the $s_0$ dependence of a single $A^{(2,m)}(s_0)$ moment, one can extract the values of $\alpha_s(m_\tau^2)$ and the power corrections $\mathcal{O}_{2(m+2)}$ and $\mathcal{O}_{2(m+3)}$. Fitting the $V+A$ distribution in a range of $s_0$ above some $\hat{s}_0 \ge 2.0$ GeV$^2$, one finds a quite poor sensitivity to power corrections, as expected, but a surprising stability in the extracted values of $\alpha_s(m_\tau^2)$ at different $\hat{s}_0$. Including the information from three different moments ($m = 0, 1, 2$), and adding as an additional theoretical error the fluctuations with the number of fitted bins, one gets the $\alpha_s(m_\tau^2)$ values given in the fourth line of Table 2. This determination is much more sensitive to potential violations of quark-hadron duality because the $s_0$ dependence of consecutive bins feels the local structure of the spectral function. The agreement with the determinations in the first three lines of Table 2 corroborates the small size of duality-violation effects in the fitted region, thanks to their very efficient suppression in the doubly-pinched moments $A^{(2,m)}(s_0)$ and the flat shape of the $V+A$ hadronic distribution above $s_0 = 2.0$ GeV$^2$.

A different sensitivity to power corrections and duality-violation effects can be achieved

with the exponentially-suppressed moments $\omega_a^{(1,m)}(x) = (1-x^{m+1})\,e^{-ax}$ that nullify the region of high invariant-mass values, strongly reducing any violations of quark-hadron duality, at the price of being exposed to all condensates. The OPE corrections become independent of $m$ when $a \gg 1$, while at $a = 0$ one recovers the non-exponential moments $A^{(1,m)}(s_0)$ that are only affected by $\mathcal{O}_{2(m+2)}$. Thus, in a purely perturbative determination of the strong coupling, the neglected OPE corrections should manifest in a larger instability under variations of $s_0$ at $a \neq 0$. Moreover, at fixed $s_0$ the splitting among moments should increase with the Borel parameter $a$, before converging at $a \to \infty$. However, the detailed analysis performed in Ref. [16], with seven different $\omega_a^{(1,m)}(x)$ moments ($0 \leq m \leq 6$), finds stable results for a broad range of values of both $s_0$ and $a$, where the power corrections do not appear to be numerically relevant. Including the information from all moments, one gets the values of $\alpha_s(m_\tau^2)$ in the fifth line of Table 2.

## 6 Summary

Table 2 displays a very consistent set of results, obtained with different numerical approaches that have different sensitivities to potential non-perturbative corrections. They are rooted in solid theoretical principles and exhibit a good stability under small variations of the fit procedures. The excellent overall agreement, and the many complementary tests successfully performed, demonstrate their robustness and reliability. Combining the CIPT and FOPT averages, we get our final determination of the strong coupling

$$\alpha_s^{(n_f=3)}(m_\tau^2) = 0.328 \pm 0.013\,. \tag{10}$$

After evolution up to the scale $M_Z$, the strong coupling decreases to

$$\alpha_s^{(n_f=5)}(M_Z^2) = 0.1197 \pm 0.0015\,, \tag{11}$$

in excellent agreement with the direct measurement at the $Z$ peak from the $Z$ hadronic width, $\alpha_s(M_Z^2) = 0.1196 \pm 0.0030$ [25]. The comparison of these two determinations, performed at completely different energy scales, provides a beautiful test of the predicted QCD running:

$$\alpha_s^{(n_f=5)}(M_Z^2)\Big|_\tau - \alpha_s^{(n_f=5)}(M_Z^2)\Big|_Z = 0.0001 \pm 0.0015_\tau \pm 0.0030_Z\,. \tag{12}$$

The $\alpha_s(m_\tau^2)$ determination could benefit from future high-precision measurements of the $\tau$ spectral functions, specially in the higher kinematically-allowed energy bins. An improved understanding of higher-order perturbative corrections is also needed in order to improve the theoretical accuracy.

## Acknowledgements

I would like to thank the local organizers for making possible this successful workshop. I want also to thank Antonio Rodríguez-Sánchez for a very enjoyable collaboration and for his many enlightening comments on the content of this manuscript. This work has been supported in part by the Spanish State Research Agency and ERDF funds from the EU Commission [Grants FPA2017-84445-P and FPA2014-53631-C2-1-P], by Generalitat Valenciana [Grant Prometeo/2017/053] and by the Spanish Centro de Excelencia Severo Ochoa Programme [Grant SEV-2014-0398].

# A   Use and Misuse of Spectral Ansatzs

Ref. [24] advocates to use observables that maximize the violations of quark-hadron duality. While this could be an interesting approach to study those uncontrollable effects, it is not the right way to perform a precise determination of the QCD coupling. The duality-violation correction to a given moment is estimated with the formal identity [26–29]

$$\frac{i}{2} \oint_{|s|=s_0} \frac{ds}{s_0} \omega(s) \left\{ \Pi_{V/A}(s) - \Pi_{V/A}^{\text{OPE}}(s) \right\} = -\pi \int_{s_0}^{\infty} \frac{ds}{s_0} \omega(s) \Delta\rho_{V/A}^{\text{DV}}(s). \tag{13}$$

The differences $\Delta\rho_{V/A}^{\text{DV}}(s)$ between the physical spectral functions and their OPE approximations are first parametrized through a functional ansatz to be fitted with the available low-energy data. This parametrization is then used to estimate the right-hand-side integral in Eq. (13), which unavoidably introduces some degree of model dependence.

We can consider the slightly generalized ansatz (in GeV units)

$$\Delta\rho_{V/A}^{\text{DV}}(s) = s^{\lambda_{V/A}} e^{-(\delta_{V/A} + \gamma_{V/A}s)} \sin(\alpha_{V/A} + \beta_{V/A}s), \qquad s > \hat{s}_0, \tag{14}$$

which coincides with the model assumed in Ref. [24] for $\lambda_{V/A} = 0$ . This functional form is expected to reasonably describe the fall-off of duality violations at very high invariant masses. However, it is not theoretically compelling at low energies and must only be regarded as an exploratory tool.

Ref. [24] assumes the ansatz to be valid above $\hat{s}_0 \sim 1.5$ GeV$^2$ and performs a fit, bin by bin, to the $s_0$ dependence of $A_V^{(0,0)}(s_0)$, *i.e.*, to the integral of $\rho_V(s)$ without any weighting. This is equivalent to a direct fit of the vector spectral function in the interval $\hat{s}_0 < s_0 < m_\tau^2$, plus the moment $A_V^{(0,0)}(\hat{s}_0)$ at the lowest invariant-mass $\hat{s}_0$ ($\sqrt{\hat{s}_0} \sim 1.2$ GeV) [16]. This moment does not receive OPE corrections, but it is very exposed to duality violations. The fitted value of $\alpha_s$ is then mainly driven by the information at $\hat{s}_0$, the lower end of the fitted range, where perturbation theory is less reliable. The higher energy bins are used to determine the spectral ansatz parameters. The assumed parametrization modifies $A_V^{(0,0)}(\hat{s}_0)$, through Eq. (13), introducing an unwanted correlation with the extracted value of $\alpha_s$. Thus, one loses theoretical control and gets at best an effective model description with unclear relation to QCD.

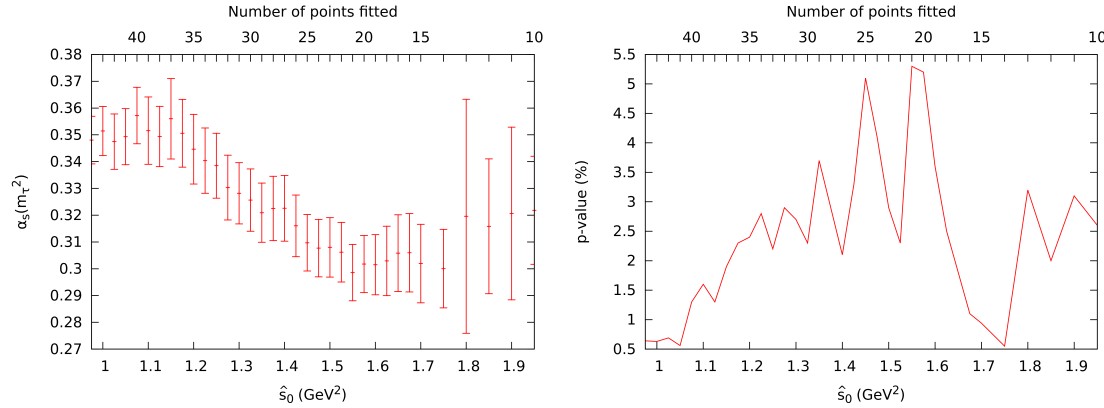

Figure 4: FOPT determination of $\alpha_s^{(n_f=3)}(m_\tau^2)$ from the $s_0$ dependence of $A_V^{(00)}(s_0)$, fitting all $s_0$ bins with $s_0 > \hat{s}_0$, as function of $\hat{s}_0$, using the approach of Ref. [24].

Taking $\lambda_V = 0$, we have reproduced the results of Ref. [24]. The left panel in Fig. 4 shows the extracted values of $\alpha_s(m_\tau^2)$ at different $\hat{s}_0$, with FOPT (CIPT gives a similar behaviour).

**SciPost**                                                    SciPost Phys. Proc. 1, 036 (2019)

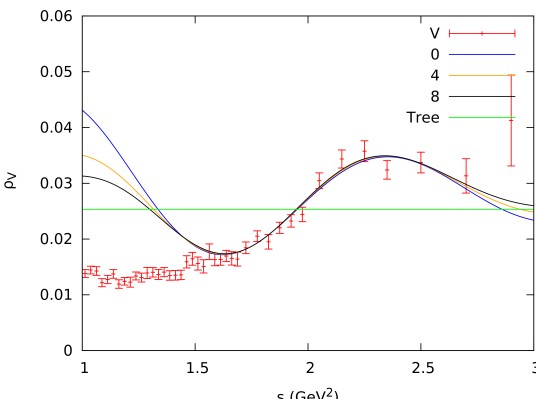

Figure 5: Vector spectral function $\rho_V(s)$, fitted above 1.55 GeV$^2$ with the ansatz (14), for different values of $\lambda_V = 0, 4, 8$, compared with the data points [16]

Table 3: Fitted values of $\alpha_s^{(n_f=3)}(m_\tau^2)$, in FOPT, and the spectral ansatz parameters in Eq. (14) with $\hat{s}_0 = 1.55$ GeV$^2$, for different values of the power $\lambda_V$ [16]

| $\lambda_V$ | $\alpha_s^{(n_f=3)}(m_\tau^2)$ | $\delta_V$ | $\gamma_V$ | $\alpha_V$ | $\beta_V$ | p-value (% ) |
|---|---|---|---|---|---|---|
| 0 | $0.298 \pm 0.010$ | $3.6 \pm 0.5$ | $0.6 \pm 0.3$ | $-2.3 \pm 0.9$ | $4.3 \pm 0.5$ | 5.3 |
| 1 | $0.300 \pm 0.012$ | $3.3 \pm 0.5$ | $1.1 \pm 0.3$ | $-2.2 \pm 1.0$ | $4.2 \pm 0.5$ | 5.7 |
| 2 | $0.302 \pm 0.011$ | $2.9 \pm 0.5$ | $1.6 \pm 0.3$ | $-2.2 \pm 0.9$ | $4.2 \pm 0.5$ | 6.0 |
| 4 | $0.306 \pm 0.013$ | $2.3 \pm 0.5$ | $2.6 \pm 0.3$ | $-1.9 \pm 0.9$ | $4.1 \pm 0.5$ | 6.6 |
| 8 | $0.314 \pm 0.015$ | $1.0 \pm 0.5$ | $4.6 \pm 0.3$ | $-1.5 \pm 1.1$ | $3.9 \pm 0.6$ | 7.7 |

The p-values of the different fits, given in the right, have a very poor statistical quality for all $\hat{s}_0$ values. The $\alpha_s$ determination of Ref. [24] is just taken from the point $\hat{s}_0 = 1.55$ GeV$^2$, where $\alpha_s(m_\tau^2)$ is smaller, with the argument that it has the larger (but still too small) p-value.[1] This procedure does not have any solid justification.[2] The p-value falls dramatically when one moves from this magic point, becoming worse at higher $\hat{s}_0$ where the model should work better. The impact of this unaccounted systematic uncertainty on the strong coupling becomes clear when one observes that the fitted value of $\alpha_s(m_\tau^2)$ is very unstable under small variations of $\hat{s}_0$. Just removing from the fit one of the 20 fitted points, results in fluctuations of the order of $1\sigma$.

The assumed model strongly deviates from the data, outside the region where the spectral function has been fitted. Fig. 5 compares the experimental spectral function with the fitted ansatz, for three different values of $\lambda_V = 0, 4, 8$. All models reproduce well $\rho_V(s)$ in the fitted region ($s \geq 1.55$ GeV$^2$), but they fail badly below it. The worse behaviour is obtained with the default model ($\lambda_V = 0$) assumed in Ref. [24]. Increasing $\lambda_V$, the ansatz slightly approaches the data below the fitted range, while the exponential parameters $\delta_V$ and $\gamma_V$ adapt themselves to compensate the growing at high values of $s$ with the net result of a smaller duality-violation correction. The statistical quality of the fit improves also with growing values of $\lambda_V$, as shown in Table 3 that gives the fitted parameters for different models ($0 \leq \lambda_V \leq 8$), taking always the reference point $\hat{s}_0 = 1.55$ GeV$^2$.

Table 3 exhibits a strong correlation between $\alpha_s(m_\tau^2)$ and the assumed ansatz. Since we are

---

[1]Using the same prescription in the axial channel, we find a better local maximum ($p = 16\%$) for $\hat{s}_0 = 1.3$ GeV$^2$ that corresponds to $\alpha_s^{(n_f=3)}(m_\tau^2) = 0.332 \pm 0.011$ (FOPT) [30,31].

[2]If the bad convergence to the data below $\hat{s}_0$ is ignored, one can find sets of model parameters (with $\lambda_V = 0$) at higher $\hat{s}_0$ that give better fits (p-values) with completely different values of $\alpha_s$ [30,31].

just fitting models to data without any solid theoretical basis (the OPE is not valid in the real axis), the strong coupling has been converted into one additional model parameter. In spite of all caveats, one gets still quite reasonable values of $\alpha_s$, but the actual uncertainties are much larger than the very naive fit errors shown in the table, which totally ignore the strong instabilities appearing as soon as one moves from the selected point $\hat{s}_0 = 1.55$ GeV$^2$. For the default $\lambda_V = 0$ model, for instance, the fluctuations of $\alpha_s(m_\tau^2)$ in the interval $\hat{s}_0 \in [1.15, 1.75]$ GeV$^2$ increase by a factor of three the error quoted in Table 3 [16]. As the fit quality improves with growing values of $\lambda_V$, the fitted central values of $\alpha_s(m_\tau^2)$ approach also the much more solid determinations quoted in Table 2.

Thus, the fitted values of $\alpha_s(m_\tau^2)$ obtained with this method strongly depend on the assumed spectral function model and, therefore, are unreliable. The claimed result in Ref. [24] is just a consequence of the particular choice of model adopted and the quoted uncertainties are largely underestimated.

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
