# Peer review of "Tau-decay determination of the strong coupling"

_SciPost Physics Proceedings, doi:SciPost Phys. Proc. 1, 036 (2019)_

## Round 1 · Referee Report · Anonymous (Referee 1) · 2018-12-9

Strengths
- The analysis reported in this paper is based on convincing scientific arguments and the strategy is well thought.
- The paper is very nicely written, and all relevant aspects are clearly discussed.
- I find the analysis in the Appendix of crucial importance in order to assess the validity of recent claims, Ref [24] in this paper, on the importance of duality violations in tau-decay determinations of the QCD coupling. In fact, the appendix shows clearly the limitations of the model-dependent description of the duality violations when the analysis is extended to larger ranges of $s$ values.
Report
This contribution is very clearly written and addresses with solid scientific arguments a relevant topic that has recently been the subject of lively debates among phenomenologists.
I am of the opinion that this debate must be resolved soon, in order to avoid generating further confusion in the scientific community, and that it can be resolved on the basis of current evidences, like the ones reported in this contribution.
I am of the opinion that this debate must be resolved soon, in order to avoid generating further confusion in the scientific community, and that it can be resolved on the basis of current evidences, like the ones reported in this contribution.
Requested changes
No changes are requested.

---

## Editorial Decision

published